# New Insight on Photoisomerization Kinetics of Photo-Switchable Thin Films Based on Azobenzene/Graphene Hybrid Additives in Polyethylene Oxide

**DOI:** 10.3390/polym12122954

**Published:** 2020-12-10

**Authors:** Qais M. Al-Bataineh, Ahmad A. Ahmad, Ahmad M. Alsaad, Ahmad Telfah

**Affiliations:** 1Department of Physics, Jordan University of Science & Technology, P.O. Box 3030, Irbid 22110, Jordan; qalbataineh@ymail.com (Q.M.A.-B.); alsaad11@just.edu.jo (A.M.A.); 2Leibniz Institut für Analytische Wissenschaften-ISAS-e.V., Bunsen-Kirchhoff-Straße 11, 44139 Dortmund, Germany; telfah.ahmad@isas.de; 3Hamdi Mango Center for Scientific Research (HMCSR), the Jordan University, Amman 11942, Jordan

**Keywords:** polymerized nanocomposite thin films, photoisomerization processes, trans-cis-isomers, molecular solar thermal energy storage media, photo-switchable thin films

## Abstract

In this work, we reported a new insight on the kinetics of photoisomerization and time evolution of hybrid thin films considering the azo-dye methyl red (MR) incorporated with graphene accommodated in polyethylene oxide (PEO). The kinetics of photoisomerization and time-evolution of hybrid thin films were investigated using UV-Vis s and FTIR spectroscopies, as well as appropriate models developed with new analytical methods. The existence of azo-dye MR in the complex is crucial for the resource action of the *trans*
↔
*cis* cycles through UV-illumination ↔ Visible-illumination relaxations. The results of the UV–Vis and the FTIR investigations prove the cyclical *trans*
 ↔ 
*cis-*states. Consequently, PEO-(MR-Graphene) hybrid composite thin films can be introduced as possible applicants for photochromic molecular switches, light-gated transistors, and molecular solar thermal energy storage media.

## 1. Introduction

Photo-switchable thin films can be isomerized between two metastable states through light-illumination. This type of thin-film has gained noticeable attention in many applications in physics, chemistry, and biology. The striking feature of photo-switchable thin films is that the two isomers have different physical and chemical properties. Recently, the azobenzene (AZO) has gained great momentum owing to its potential applications in photochromic molecular switches [1], light-gated transistors [2], and molecular solar thermal energy storage media [3,4,5,6]. AZO has two geometric forms, *trans-*state and *cis*-state. All AZO thin films are initially in the *trans*- isomerization state since it is thermally stable at room temperature [7,8]. The *trans*
*→ cis* isomerization may occur by exposing thin films to the UV-light illumination, while the reverse *cis* → *trans* isomerization can occur by either illuminating thin films with visible light (400–450 nm) or by thermal excitation in a dark environment [9,10]. The photoisomerization mechanisms of the *trans*↔*cis* isomers in the blended polymers play a key role in extracting the proper chemo physical properties of additive azo benzene [7,10,11,12,13].

Polymers go under a photodegradation process via UV light-absorption (in particular) by backbone-carbonyl groups induced by photochemical reactions. Photodegradation of polymers usually occurs through either chain scission (reduction in molecular weight) formatting C=C double bonds, cross-linking, and hydroperoxide O–H reactions that activate the polymer-molecule while absorbing light. The degradation process begins with light-absorption by a photo-initiator, then photo-cleavage of polymeric-molecules, transferring them into free-radicals that further enhances the degradation process. The level of degradation (breakage) is relevant to the light-energy used. Shorter light-wavelengths do not need oxygen for the hydroperoxide process, while low-wavelength sources need an oxygen environment [14,15,16,17]. Azo dyes usually degrade under the influence of UV-irradiation, especially when a photocatalytic source of metal oxide is presented. ZnO, TiO2, SnO2, and CuO are mostly used as photocatalytic oxidation sources for the degradation process [18,19]. Methyl red is of those azo dyes which contain one or more azo groups (–N=N–) as a chromophore group that is influenced by UV absorption. Recently, 2D carbon as graphene and graphene oxides were used as support materials for photocatalytic processes [20,21].

The mutuality of the exceptional transformation of the photoisomerization processes via thermally relaxed stable *trans*-isomerization and nonstable *cis*-isomerization via illumination has become of the main theoretical and practical perspectives [22,23,24]. Generally, the photoisomerization process exhibits several mutual operations involving bending or non-bending aromatic rings in micromolecular composite chains [25,26].

We selected polyethylene oxide (PEO) as a host polymer owing to its semi-crystallinity nature with two phases of crystallite and amorphous forms coexisting at normal conditions [27] and its low absorbance values compared to the other host polymers [28,29,30]. Graphene has flattened sp^2^-crossed networks with π-electron restrained over the rings [31]. Graphene integrates groups of carboxylic, carbonyl, and hydroxyl assemblies at the boundaries, including hydroxyl and epoxy assemblies at the basal planes. In addition, the aromatic AZO acts as a frail electrophile and outbreaks carbon atoms in a dense electron fog form in a hydroxybenzene ring [32]. One of the important benefits of selecting graphene in the composite is its behavior as a photo-initiator [33].

This work’s main objective is to investigate the kinetics of photoisomerization processes of polymeric thin films performed with azo-dye (methyl red) incorporated with graphene accommodated in polyethylene oxide (PEO). Understanding the mechanism of photoisomerization is crucial for the practical implementation of this nanocomposite in device fabrication. The optical and chemical characteristics of the *trans*- and *cis*-isomers of the PEO-MR-Graphene hybrid complexes are usually investigated via the UV–Vis and the FTIR techniques. To the best of our knowledge, we are not aware of any practical investigation of the kinetics of PEO-MR-Graphene hybrid composites’ methodological isomerization via optical studies.

## 2. Experimental Details

### 2.1. Materials

Polyethylene Oxide (PEO) (–CH_2_CH_2_O–)_n_ with a molecular weight of 300,000 g/mol and graphene powder (<20 µm) with a molecular weight of 12.01 g/mol were obtained from Sigma-Aldrich Co. Inc: Munich, Germany Methyl-Red (MR) (C_15_H_15_N_3_O_3_) of pH level between 4.2 and 4.6 powder supplied by SCP SCIENCE: Montreal, QC, Canada.

### 2.2. Synthesis of PEO-MR-Graphene Hybrid Composite Thin Films

All solutions were prepared using absolute methanol (CH_3_OH, with a purity of 99.8%). One gram of PEO was successively dissolved in 100 mL methanol at 45 °C by using continuous magnetic stirring for 5 h. MR-Graphene solution (1 mole: 1 mole) was mixed via solid-state blending using an agate mortar (BUCHI™ Achat Mörser mit Pistill: Fisher scientific, Hampton, NH, United States). After that, one gram of MR-Graphene mixture was dissolved in 100 mL methanol at room temperature. The hybrid solution of PEO-(MR-Graphene)/methanol was obtained by mixing PEO/methanol and MR-Graphene/methanol in a (3 to 1) ratio via stirring them magnetically for a duration of 6 h. The composites were homogenized via sonication for 6 h. The final solution was treated with a 0.45 μm Millipore filter. The hybrid thin films with 300 nm thickness (measured in cross-sectional view using a SEM micrograph: Fisher scientific, Hampton, NH, United States) were dip-coated for one hour on glass substrates. The solvents were evaporated, and the organic residues were removed by air drying the films at 70 °C for half an hour.

### 2.3. Characterization

The absorbance measurements were performed by a Double-Beam UV–vis Spectrophotometer (U-3900H:Hitachi, Tokyo, Japan) at room temperature. The hybrid nanocomposite vibrational bands were obtained using the Fourier transform infrared spectroscopy (FTIR) (Bruker Tensor 27 spectrometer with a disc of KBr: Billerica, MA, United States spectrometer with a disc of KBr). FTIR measurements were performed by pealing-off the films out of the glass substrates and used as a solid form in the system. The thermal stability was investigated using the Thermogravimetric Analysis (TGA) technique.

The PEO-(MR-Graphene) thin-film were illuminated by a UV-light source with 366 nm wavelength and 6 Watts of power (32 W/cm^2^ of intensity) for 0, 30, 60, 120, 240, and 480 s in order to investigate the impact of the time-evolved in transforming the *trans*-isomerization phase to *cis*-phase. The absorbance spectra were measured at every UV-illumination time-exposure. The films were exposed to blue light-illumination by a visible source with a wavelength of 467 nm and power of 6 Watts (32 W/cm^2^ of intensity) for 0, 90, 180, 360, 720, 1440 s. The process was done to inspect the influence of the exposure-time on the reversal of the monomer *cis*-isomerization state back to the original *trans*-state. The absorbance spectra were measured at every exposure-time evolved by the visible-illumination. Moreover, the FTIR spectroscopy measurements were employed as additional supportive evidence of the influence of the UV-Vis illuminations via exploring the vibrational changes that occurred in the bonding modes of the *trans*- and the *cis*-cases, accordingly.

## 3. Results and Discussion

### 3.1. Characterization of PEO-(MR-Graphene) Hybrid Nanocomposite Thin Film

As previously reported [3,33], the chemical structure of the AZO-Graphene used in this work can be visualized, as shown in Figure 1a. The coupling agent forming the AZO-Graphene hybrid contains the AZO-nitrogen atom bonded covalently with the carbon atom that is also bonded with the graphene aromatic ring. The chemical formula of PEO is C_2n_H_4n+2_O_n+1,_ and that of MR is given by C_15_H_15_N_3_O_3_. The covalent linkage between methyl red and Graphene exterior superficial shells can be determined by FTIR patterns shown in Figure 1b. The vibrational bands for graphene located at 2015 cm^−1^ and 2161 cm^−1^ represent C=C and C=O. FTIR spectrum for PEO exhibited vibrational bands between 700–1000 cm^−1^ representing bending C–H. Vibrational bands between 1000–1400 cm^–1^ are assigned to C–O stretching vibration, a band at 1470 cm^–1^ that could be ascribed to the –CH_2_ bending vibration, while a band appearing at the 2886 cm^–1^ could be ascribed to the symmetric and asymmetric C–H stretching modes of the CH_2_ group. Moreover, the significant pronounced vibrational bonds that appeared in PEO-(MR–Graphene) hybrid nanocomposite thin films were found to be related to the aromatic rings, the azo- chromophores (–N=N–), the stretching bonds of C–N, and others such as C–H. The in-plane C–H vibrational peak was found at 1122 cm^−1^, and the out-of-plane vibrational peaks were found between 1000–700 cm^−1^. The aromatic bands were observed between 1600 and 1400 cm^−1^. Moreover, the spectral peak observed at 1342 cm^−1^ was apportioned to the C–O stretching bond in PEO. According to W. Pang et al. study [3], the absorption due to the C–N bonding is unlike the C=C bonding, and its frequency-range is centered at 1342 cm^−1^ caused by the influence of resonance which upsurges the bond-order assigned to that particular ring in the chain and the dangling N-atom. The presence of C–N bonds designates the covalent bonding between MR to the graphene-outer surface. The peak observed at 3732 cm^−1^ was assigned to the O–H bond in PEO. However, the peak at 2861 cm^−1^ was dispensed to C–H stretching. Moreover, the peak found at 2114 cm^−1^ was assigned to double bonding C=C. The peaks observed at 1692 cm^−1^ and 1583 cm^−1^ were related to the double bonding of C=O. Finally, the peak observed at 3732 cm^−1^ was consigned to the single bonding of O–H.

Figure 1c shows PEO and PEO’s absorbance spectra-(MR–Graphene) hybrid composite thin films as a function of wavelength. The PEO absorption spectra-(MR–Graphene) hybrid composite is characterized by a band transition. Namely, the $\pi -\pi^{*}$ in the range of 350–650 nm. However, PEO thin film does not contain any absorption band identified in this region [9]. Clearly, the absorbance of the PEO-(MR–Graphene) hybrid composite thin film exhibits $\pi -\pi^{*}$ transition in the (350–650) nm range with a maximum at 422.5 nm. This spectral behavior is believed to occur due to the presence of MR in the film nanocomposite giving the films the red-appearance color. Hence, the films show high absorbance for green, blue, and violet-colored light [7,10,11,12,13].

Furthermore, PEO’s thermal stability-(MR–Graphene) hybrid nanocomposites were investigated using the thermogravimetric analysis (TGA) technique at temperatures up to 400 °C, as shown in Figure 1d. The PEO-(MR–Graphene) hybrid nanocomposite shows a stable weight-loss starting at the ambient temperature T and extends to 150 °C, at which the mass-loss becomes 8% approximately. The mass-loss was considered as the loss of water/solvents adsorbed in the sample. The mass-loss curve versus temperature then sharply declined in the temperature interval ranging between 150 to 200 °C. Within this interval, the mass-loss is significant (80%), indicating the influence of the intermolecular/intramolecular bonding change when the nanocomposite is exposed to high temperatures. PEO’s thermal stability-(MR-Graphene) hybrid nanocomposite is lower than that of PEO, as shown in Figure 1d. For PEO, the mass-loss versus temperature drops sharply at around 300 °C [34,35]. Interestingly, despite the slight negligible slope in the TGA relation below 150 °C, TGA thermograms confirm that the PEO-(MR–Graphene) hybrid composite was stable below 150 °C. Most of the practical optical applications are feasible below 150 °C.

### 3.2. Kinetics of Photoisomerization Processes

In this section, we investigated and reported the kinetics of PEO transformation-(MR–Graphene) hybrid nanocomposite thin films from the original *trans*-state to *cis*-state using UV-exposure. The reversed transformation to *trans*-state via visible-illumination was also discussed and interpreted. UV–light primarily illuminates the PEO-(MR–Graphene) hybrid nanocomposite thin films for a certain period of time. The films are then exposed to Blue-light illumination for another period. Figure 2a illustrates the absorbance spectra of the PEO-(MR–Graphene) hybrid nanocomposite thin films illuminated by UV-light for various periods. The major absorption peak of PEO-(MR–Graphene) at the initial *trans*-state in the visible range was found to be at 422.5 nm with an absorbance amplitude of 0.169%. The film was then exposed to UV light of 366 nm for 30, 60, 120, 240, and 480 s, respectively. Moreover, the films show a variant absorbance band in the middle of the visible region (380–625 nm) with a peak-blue-shifted and amplitude-decrease transferring the *trans*-state to a *cis*-state, as expected. The photoisomerization process is not a spontaneous instantaneous process. Instead, it occurs through steadily building up sub-stages. The film exposure to the UV-illumination reveals a noticeable reduction in the absorbance at ~422.5 nm in the *trans*-isomerization stage and the presence of a dual peak at ~470 nm created from the *cis*-isomerization state. A steady reduction in the $\pi –\pi^{*}$ electron transition band of the *trans*-state upon the increasing of UV-illumination time duration until a photostationary equilibrium state between the *trans-* and the *cis-*isomers was detected. The absorbance spectra of PEO-(MR–Graphene) thin films prior-to and in post-UV illumination revealed the existence of dual isosbestic figure-tips that appeared close to 380 and 625 nm, respectively. Moreover, the UV-illumination of PEO-(MR–Graphene) hybrid nanocomposite thin film leads to the transformation from *trans-*state to a *cis-*state. In other words, the absorbance peak at wavelengths ranging from 380 nm to 625 nm decreased as the period of illumination increased. The prolonged illumination does not lead to any changes in the absorption spectra, confirming that a quasi-state of photo-stationary phase occurs between the *trans-* and the *cis-*stats. The long-time isomerization (480 s of illumination) is needed until a photo-stationary equilibrium is achieved, compared with previous results reported [7,9,10]. The observed electronic transitions in the visible range indicated a suitably effective absorption of solar power with high-energy. The solar energy absorbed in the material is of prime necessity to achieve two- or three-multiples of energy more than the activation barrier-energy $\Delta E_{i}$ (needed for complete isomerization of AZO-molecules) [36]. Based on the findings in the literature, the energy reserved/AZO-molecules (ΔH) for *cis*-isomerization has to be less than $\Delta E_{i}$. In other words, $\Delta H=\Delta E_{i}-\Delta E_{a}$, where, $\Delta E_{i}$ is the activation barrier (energy needed to isomerize one AZO molecule), $\Delta E_{a}$ is the thermal/optical activation barrier and efficiency of storing energy (the energy stored per the solar energy absorbed) is normally <30% [36,37].

The kinetics of the *trans-cis* photoisomerization of PEO-(MR–Graphene) hybrid nanocomposite thin films were investigated using the first-order kinetics via calculating the rate of photoisomerization, as well as the thermal/optical activation barrier $\Delta E_{a}$ for *trans-cis* photoisomerization [38,39]. The photoionization rate as a function of time $\left(p\right)$ could be written as,
(1)$$\ln\frac{A_{t}-A_{\infty }}{A_{0}-A_{\infty }}=-pt$$
where $A_{0},\;A_{t}$ and $A_{\infty }$ are the absorbance at various time conducts, namely prior to the light exposure (initial *trans*-state), during radiation in time *t*, and post to the light exposure for a protracted time. The average value of $\ln\left(A_{t}-A_{\infty }/A_{0}-A_{\infty }\right)$ of PEO-(MR–Graphene) hybrid nanocomposite thin film is plotted against the period (*t*), as shown in Figure 2b. Obviously, $\ln\left(A_{t}-A_{\infty }/A_{0}-A_{\infty }\right)-t$ relation has two distinct linarites. The discontinuity occurs at $t_{c}\;\sim \;240\;\min$, then the curve falls off with less steepness as a function of illumination time. The area’s ratio under the absorption curve in the visible range for all UV-illumination periods for the *cis*-hybrid isomerization was calculated with respect to the trans-hybrid isomerization. It was found to be around 95.5% at $t_{c}$ for the PEO-(MR–Graphene) hybrid nanocomposite thin films. The rate constant $\left(p\right)$ was found from the slope to be $p_{1}=3.822\;\times \;10^{-3}\;s^{-1}$ for $0<t<t_{c}$, $p_{2}=1.634\;\times \;10^{-4}\;s^{-1}$ for $t_{\infty }>t>t_{c}$. Moreover, the variations in p could be reasonably attributed to the disparity of $\Delta E_{a}$ [37] and calculated as,
(2)$$\Delta E_{a}=-kT\ln\frac{h\ln2}{\tau_{1/2}kT}$$
where: $\tau_{1/2}$ is the time needed to transfer half the *trans*-states into *cis-*states; $\tau_{1/2}=\ln2/p$, k is Boltzmann constant, h is Plank constant, and T is the temperature. Table 1 shows that $\tau_{1/2}^{1}=3.022\;\min$ for films illuminated for a time less than $t_{c}$, while $\tau_{1/2}^{1}=70.692\;\min$ for those films kept under irradiation for time-periods longer than $t_{c}$. Equation (2) indicates that $\Delta E_{a}=2.012\;\mathrm{eV}\;at\;0<t<t_{c}\;and=\;2.093\;\mathrm{eV}\;$ at ($t_{\infty }>t>t_{c}$), respectively.

Beyond $t_{c}$, the isomerization energy barrier is higher than that below, and the process proceeds even slower. The calculations agree well with the observed red-shift for the band $n–\pi^{*}$ in the absorption spectrum beyond $t_{c}$ where the *cis*-isomer contents have exceeded the *trans*-isomer contents in the PEO-(MR–Graphene) hybrid nanocomposite thin film. Therefore, the separation between the two neighboring *trans*-isomers gets longer, and the intermolecular interactions reduce accordingly. In this scenario, the transformation from the *trans-*to-*cis* photoisomerization is limited to its steric effect, and consequently, the reaction barriers are primarily dominated by the electronic configuration of the −N=N− group [3]. Based on the related findings in the literature [3,33], one may plot the chemical structure of AZO-Graphene used in this work, as shown in Figure 2c. The chemical structure alters as the period of UV-illumination increases leading to enhanced transformation from *trans*-states to *cis*-states. Figure 2d,e shows the *cis*-isomerization of the AZO-Graphene hybrid composite prior to and beyond the $t_{c}$ reaching the 95.5%.

Another evidence for the kinematic transformation from *trans*-state to *cis*-state for the PEO-(MR–Graphene) hybrid composite thin film is revealed by analyzing the FTIR spectrum. Figure 3 shows the FTIR spectra for the *trans-* and *cis-*states in the spectral range of 500–4000 cm^−1^. It was observed that the spectral peak for the O–H bond found at 3732 cm^−1^ in the *trans*–state had disappeared in the *cis*–state. This indicated that the free radicals generated by the films’ UV irradiation had broken the weak O–H bonds in a photoreaction process. Moreover, the transmittance spectra as a function of frequency for *cis*- and trans-MR showed nearly similar trends with small portions of the bands that were shifted to lower energies. This reflection confirmed that such interactions had a slight influence on N=N bands at frequencies between 1400–1600 cm^−1^. The evaluated intensity in the infra-red region of N=N band for the *cis*-MR was enormously high compared to that for the *trans-*MR, confirming the enhanced photoisomerization process of the PEO-(MR–Graphene) hybrid nanocomposite thin film.

Furthermore, Figure 2a indicates that each absorbance peak of the PEO-(MR–Graphene) hybrid nanocomposite thin film contains two single absorption peaks (two frequency bands). As the UV-illumination duration increased, the absorption amplitude decreased, indicating that the transformation from *trans*-state to *cis*-state had occurred accordingly. Increasing the duration of the UV-light exposure leads to developing new configured dual-shapes in the absorption peaks shown in Figure 2a, which indicates the development of an impeded dual-frequency bands forming two distinct crests instead of one main absorption peak. Additional evidence for the existence of dual overlapped sub-peaks was revealed from the clear shoulders that appeared mainly at the lower and higher frequency spectrum segments. Namely, at 470 nm and 422.5 nm, respectively.

Figure 4a–f shows the major peaks of PEO-(MR–Graphene) hybrid nanocomposite thin films for all durations of the UV-light exposure in the visible range of the spectrum as fitted to pair Gaussian crests [9]. Figure 4a describes the major peak of PEO-(MR–Graphene) hybrid nanocomposite thin films in the starting *trans*-state fitted to the dual Gaussian crests. The absorption spectra exhibited two bands (high- and low-frequency) with maxima starting at around 417.89 and 471.13 nm and line widths of 53.58 and 120.26 nm, respectively. The absorbance spectrum changed in its configured shape as the UV-illumination exposer time increased, as shown in Figure 4b–f showing the transformation from the *trans*-state to the *cis*-state.

A detailed quantitative analysis for the lower- and higher-frequency bands was performed to obtain a more in-depth insight into the influence of the UV-illumination time. A closer look at Figure 4 shows the specific detailed structure for each peak specifying the amplitude and the area variation underneath the absorbance curve as plotted in Figure 5a,b during the transformation from *trans*- to *cis*-isomerization. Each sub-band’s single configuration behavior varies with the UV-light exposure duration as generated through the systematic deviation from the line-width in the related Gaussian function. As can be seen from Figure 5a,b, the area and the amplitude of overall absorbance, low-frequency, high-frequency bands decrease continuously with time till the $t_{c}$ is reached. However, beyond the $t_{c}$, the amplitude and area under the main overall absorbance curve for the low-, high-frequency bands become constant, confirming the achievement of 95.5% proportion of the *cis*-hybrid in AZO-Graphene hybrid nanocomposite. Moreover, the high-frequency sub-band has suffered from a blue-shifting process, which indicates a bathochromic change occurred due to the light absorbed by MR. This is considered as an H-bond donor source or a dissociated-intermediate H-bond donor source [40,41]. Consequently, the overall absorption band is anticipated to occur due to the H-bond’s mutual interaction among the PEO and the azo-nitrogen contents through MR molecules. The blue-shifting for both sub-bands (low- and high-frequency bands) exhibit similar trends indicating that H-bonds have a fundamental influence on MR molecules’ various sites. Moreover, the high-frequency band demonstrates steady bathochromic shifts during the UV-illumination process, demonstrating two unambiguous symmetries at the H-bonds’ backgrounds accompanied by MR transformation. The shift in all bands’ amplitude ensures that the time-dependent photoisomerization process is achieved by more than one step [42]. This is because the photoisomerization process is a complex response to the four-level model of *trans*- and *cis*- isomerization as demonstrated by Sekkat Z. et al. [23] and Lee G.T. et al. [43], as shown in Figure 6. The four-levels model describing the *trans*- and the *cis*- isomerization has been investigated thoroughly by Al-Bataineh Q.M. [9]. As is well known, the ground state of AZO-band is a singlet (S0) while the S1 and S2 are first and second singlet excited states, respectively. The S1 state can be generated by either direct excited transition from S0 to S1 or via intersystem crossing between S1 and S2 states (i.e., relaxation of the S2 state down to S1) seen in Figure 6. The states S1 and S2 generated via trans–AZO excitations are different from those of cis–AZO excited states in their energy and conformations.

As understood from azobenzene’s behavior, it experiences a revocable photoisomerization process, i.e., a transformation from *cis*- to *trans*-isomerization. This process is usually associated with enormous quantum yields with no significant reactions in the process [44]. Figure 7a illustrates the absorption varieties of the PEO-(MR–Graphene) hybrid films at *cis*-state for numerous visible blue-light illumination durations. Beyond 480 s of UV-light exposure, the films appeared to show a *cis*-isomerization case (lowest solid yellow line). The PEO’s main absorption peak-(MR–Graphene) hybrid film in the visible region originated at 423 nm with a peak-amplitude of 0.134. The hybrid film was further exposed to the blue-visible light for 90, 180, 360, 720, and 1440 s, respectively. Figure 7a shows that the film gains various absorption bands in the intermedium portion of the visible region (380–625 nm) associated with a red-shift behavior in the crest value transporting back the film from *cis*- to the *trans*-isomerization case as predicted. We noticed that the process of transforming the material from *cis*- to *trans*-state is not a sudden spontaneous nor instantaneous transformation. However, it has been transferred via a sequence of phases. Interestingly, the kinematic process of transferring the *cis-*to-*trans* state demonstrates a kind of deformation from the first-order kinetics, as seen in Figure 7bEach hybrid form, the departure from the first-order kinetics is quite obvious. It may be assigned to the intermolecular interactions between AZO and Graphene and the modifications of AZO’s electronic structure due to the novel steric structure of the hybrid. It is clear that $\ln\left(A_{\infty }-A_{t}/A_{\infty }-A_{0}\right)-t$ has one discrete segment which behaves linearly. The rate constant *κ* could be extracted from the slope. The obtained value of $p_{1}=2.621\;\times \;10^{-3}\;s^{-1}$.

### 3.3. Investigating the Photoisomerization Cycles

Exposing the material in the *trans*-isomerization state to UV-light reveals the conversion to the *cis*-isomerization state. The new *cis*-isomer state could be converted back to the *trans*-isomerization state by either a thermal or optical relaxation process. The PEO-(MR–Graphene) hybrid films are treated with the optimum UV-illumination conditions for trans to *cis-state* and then converted back to *trans*-state via blue-light relaxation process several times. Figure 8 shows a periodically repeated photoisomerization by optical relaxation process, confirming reliable hysteresis cycles for PEO-(MR–Graphene) hybrid nanocomposite thin films with no significant characteristic-loss. This fact confirms that the PEO-(MR-graphene) hybrid composite thin films may be considered candidates for many applications such as photochromic molecular switches, light-gated transistors, and molecular solar thermal energy storage media.

### 3.4. Atomic Force Microscope (AFM) Studies

Atomic Force Microscope (AFM) measurements were performed to investigate the morphological features of the PEO-(MR–Graphene) hybrid nanocomposite thin films. Figure 9 shows the AFM before (a) and after curing (b). In both cases, films exhibit amorphous nature. Before UV curing, films’ surfaces appeared to be inhomogeneous compared with the surfaces of films after curing. Exposing films to UV curing induces strong film polymerization, yielding highly homogenous surfaces—the random distribution of the inserted MR–Graphene into PEO-host results in extremely inhomogeneous surfaces. However, illuminating films with UV light triggers MR–Graphene components of the composite to rearrange, occupying ordered sites throughout the PEO matrix. The relaxation of different composite components occurred to satisfy the minimum energy requirements and smoothen the surfaces greatly.

## 4. Conclusions

The fundamental mechanisms of the kinetics of photoisomerization of the PEO-(MR-Graphene) hybrid nanocomposite thin films are explored, evaluated, and investigated thoroughly. Deliberately, we explore and provide a new insight into the photoisomerization kinetics and time evolution for a hybrid thin-film based on Azo dye methyl red (MR) incorporated with graphene hosted in polyethylene oxide. The kinetics of photoisomerization and time evolution of the hybrid thin film was examined via the UV-Vis and the FTIR spectroscopic techniques and by using specific analytical models. The existence of the Azo Dye MR in the amalgamated composites is crucial for the effectual acts of the *trans*
↔
*cis* cyclic isomerization via UV-illumination ↔ Visible light-relaxation. Moreover, the UV–Vis and FTIR investigations confirm the hysteresis cycles of *trans-cis-*states. In conclusion, the PEO-(MR-Graphene) hybrid nanocomposite thin films are proven to be potential candidates for many applications. These applications include photochromic molecular switches, light-gated transistors, in addition to molecular solar thermal energy storage media.

## Figures and Tables

**Figure 1 polymers-12-02954-f001:**
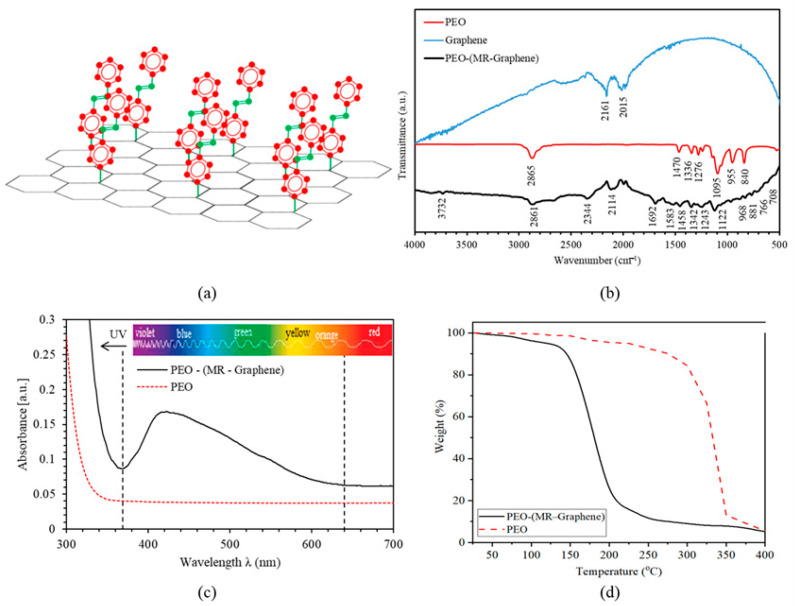
(**a**) The molecular structures of *trans*-AZO-Graphene hybrid, (**b**) FTIR spectra of PEO, Graphene and PEO-(MR–Graphene) hybrid composite thin film, (**c**) Absorbance spectra of PEO and PEO-(MR–Graphene) hybrid composite thin films versus the wavelength, (**d**) TGA curve for PEO-(MR–Graphene) hybrid composite thin film.

**Figure 2 polymers-12-02954-f002:**
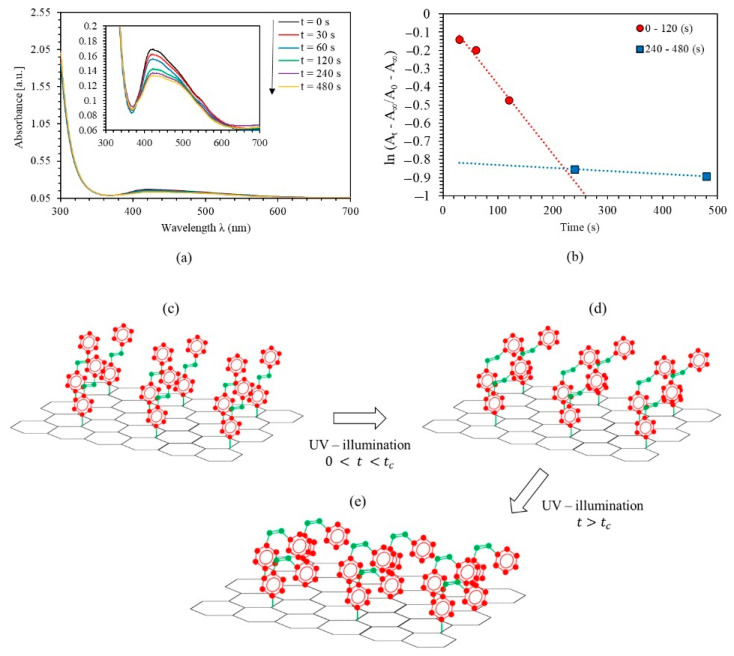
(**a**) Absorbance spectra of PEO-(MR–Graphene) hybrid composite thin film for various UV-illumination times, (**b**) the kinetic constants for *trans → cis* photoisomerization of PEO-(MR–Graphene) hybrid composite thin film, (**c**) the chemical structures for the *trans* AZO-Graphene hybrid, and (**d**,**e**) the *cis* AZO- Graphene hybrid before and after $t_{c}$.

**Figure 3 polymers-12-02954-f003:**
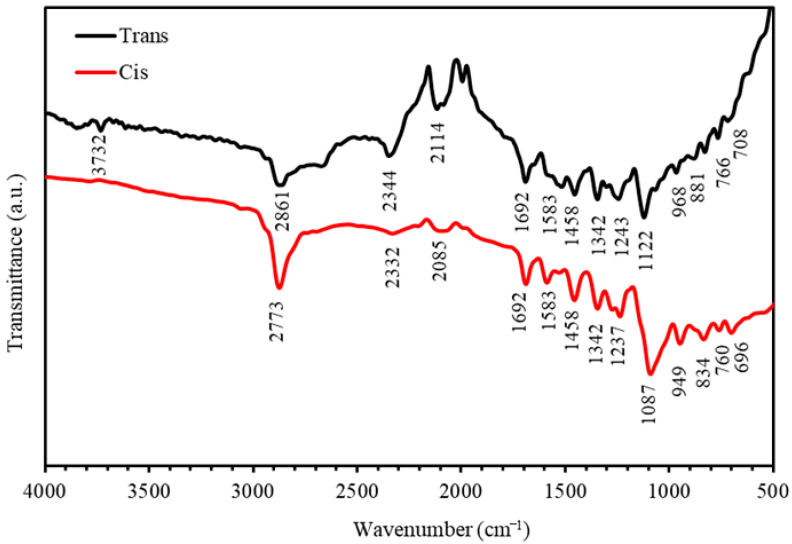
The FTIR spectra of *trans-* and *cis-*states of PEO-(MR–Graphene) hybrid composite thin films.

**Figure 4 polymers-12-02954-f004:**
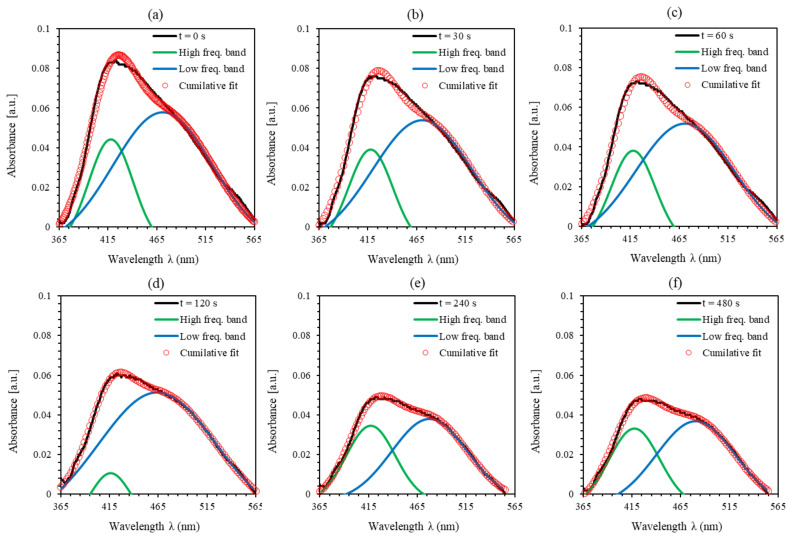
Double peaks fit for the absorbance spectra of PEO-(MR–Graphene) hybrid nanocomposite thin film for various UV-illumination exposure periods (**a**) 0 s, (**b**) 30 s, (**c**) 60 s, (**d**) 120 s, (**e**) 240 s and (**f**) 480 s.

**Figure 5 polymers-12-02954-f005:**
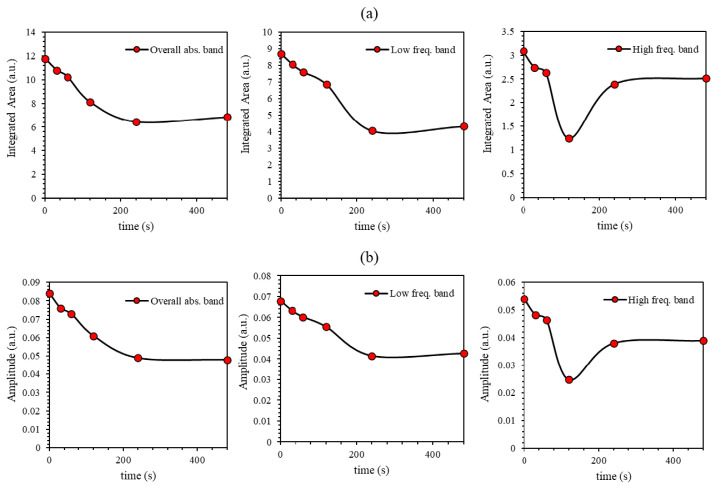
(**a**) The integrated area and (**b**) the amplitude-fit for the dual-peaks of the absorbance for the PEO-(MR–Graphene) hybrid nanocomposite thin film for various UV-light exposure periods.

**Figure 6 polymers-12-02954-f006:**
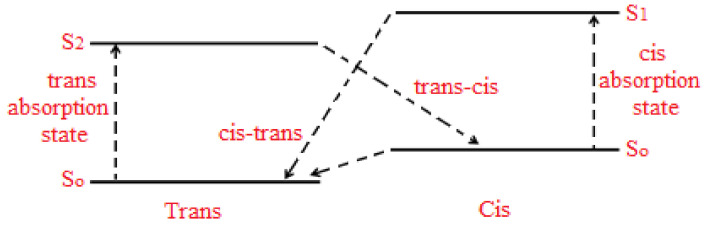
Four-levels diagram illustrating the *trans*- and the *cis*-isomerization.

**Figure 7 polymers-12-02954-f007:**
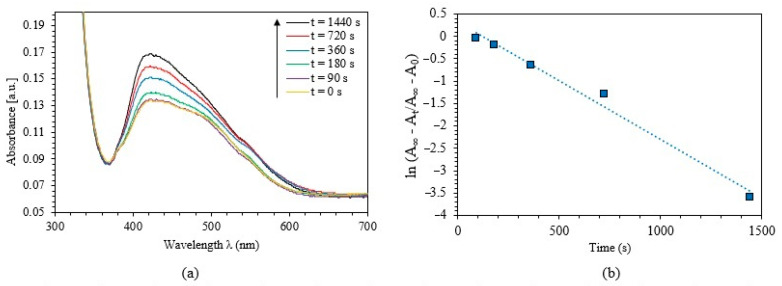
(**a**) Absorbance spectra of PEO-(MR–Graphene) hybrid composite thin film for different blue-light exposure durations, (**b**) The kinetics constants for *cis → trans* photoisomerization of PEO-(MR–Graphene) hybrid nanocomposite thin film.

**Figure 8 polymers-12-02954-f008:**
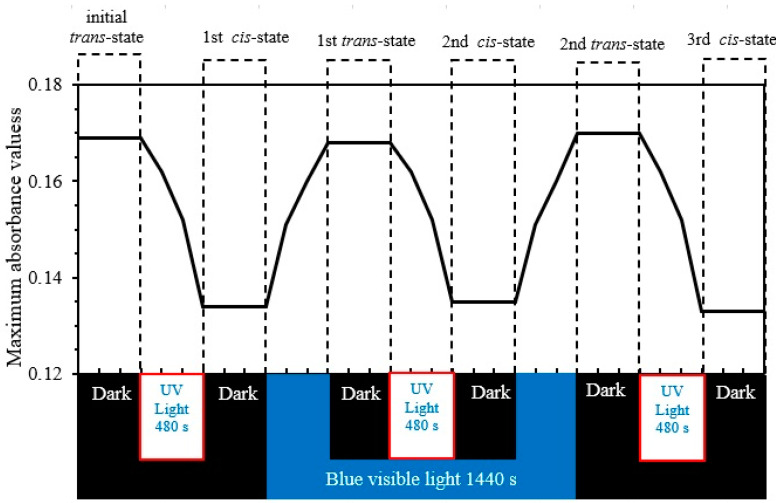
Maximum absorbance values through photochemical and thermal processes that transform from *trans* to *cis* and vice versa of PEO-(MR–Graphene) hybrid nanocomposite thin films.

**Figure 9 polymers-12-02954-f009:**
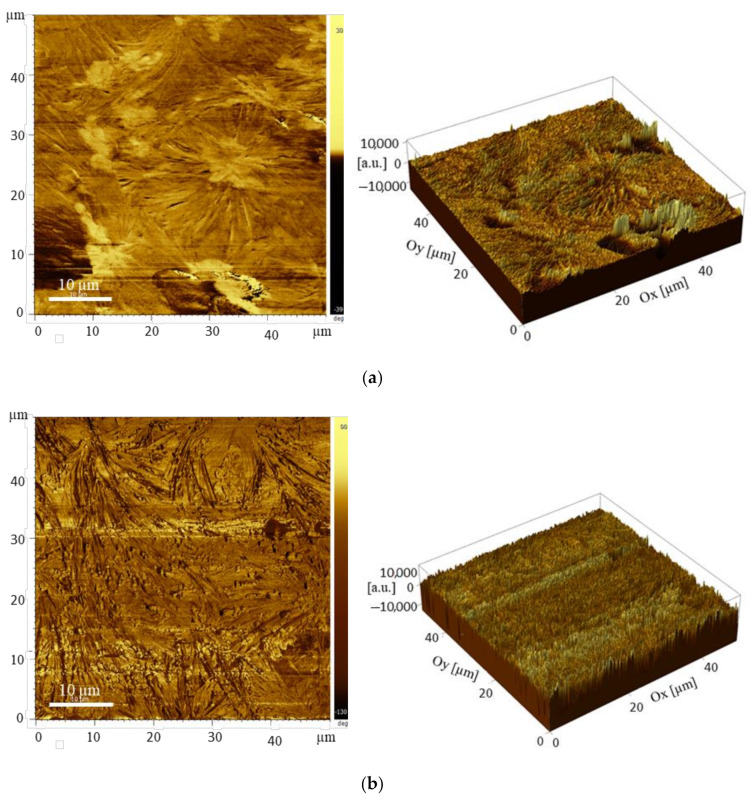
2D and 3D AFM images of PEO-(MR–Graphene) hybrid nanocomposite thin films: (**a**) without curing and (**b**) with curing.

**Table 1 polymers-12-02954-t001:** Kinetic constant (p) and thermal energy barrier ($\Delta E_{a}$) of PEO-(MR–Graphene) hybrid composite thin film ($t_{c}=240\;s$).

	$0<t<t_{c}$			$t_{\infty }>t>t_{c}$		
	$p_{1}\;(s{-}^{1})$	$\tau_{1/2}^{1}\;(\min)$	$∆E_{a}\;(\mathrm{eV})$	$p_{2}\;(s{-}^{1})$	$\tau_{1/2}^{2}\;(\min)$	$∆E_{a}\;(\mathrm{eV})$
AZO–Graphene	3.822 × 10^−3^	3.022	2.012	1.634 × 10^−4^	70.692	2.093

## Data Availability

All the data provided in this manuscript are either collected from out apparatus or derived from the raw data collected. They are available at any time for readers.

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
