# Peer review of "New Insight on Photoisomerization Kinetics of Photo-Switchable Thin Films Based on Azobenzene/Graphene Hybrid Additives in Polyethylene Oxide"

_polymers, 2020, doi:10.3390/polym12122954_

Round 1

Reviewer 1 Report

In this study, the authors investigated the photoisomerization processes of polymeric thin films with azo-dye and graphene in PEO. This article can be reconsidered for publication after a revision process, specific commentaries provided below:  

1) The manuscript contains typos (Page 1, line 33 [tans should be trans], and few others), thus, it should be rechecked one more time.  

2) What was the morphology of prepared films. Supply AFM data for the visualization. Please also provide data on how the thickness of the films was measured.

3) What was the size of graphene powder? Please supply additional evidence of graphene presence in final films (Raman or XPS). 

4) Azo dyes as well as polymers usually degraded under UV radiation. Please comment on this issue. 

5) No statistical information was found in the manuscript. How many samples per each trial were tested, mean values + SD bar charts should be provided.  

Author Response

Reviewer #1

1) The manuscript contains typos (Page 1, line 33 [tans should be trans], and few others), thus, it should be rechecked one more time.  

Reply: All Misprints in the manuscript were corrected as the reviewer suggesting. Please see the revised version.

2) What was the morphology of prepared films. Supply AFM data for the visualization. Please also provide data on how the thickness of the films was measured.

Reply: Authors added the AFM images to show the morphological properties for the films based on the reviewer suggestion. Please see page 11, lines 345-356 of the revised manuscript.

The thickness was measured using SEM micrograph, and the authors added this note to the manuscript without including the figure shown below. The figure below shows the average thickness of the films, but we see there is no need to put it in the manuscript. The authors added this into manuscript Page 2, line 88.

Note Please: See the attached file for the missing part of a figure showing the cross-sectional SEM micrograph for how we found the thin film thickness. 

3) What was the size of graphene powder? Please supply additional evidence of graphene presence in final films (Raman or XPS). 

Reply: Graphene powder is < 20 µm as supplied by Sigma Aldrich Co. Inc. The authors added this into manuscript Page 2, line 76.

Unfortunately, there are no Raman or XPS spectroscopy measurements in our hands. However, we added the FTIR spectrum of PEO and Graphene to show the presence of graphene in the films. The authors added this into manuscript Page 3, lines 115 – 121 along with the few lines after for PEO. See also Fig. 1(b).

4) Azo dyes as well as polymers usually degraded under UV radiation. Please comment on this issue. 

Reply: A short paragraph on the huge subject of UV degradation for azo dye and polymers has been added to the introduction part of the manuscript (lines 39 – 52) and References 14- 21 have been added for further background resources are suggested to the readers.

5) No statistical information was found in the manuscript. How many samples per each trial were tested, mean values + SD bar charts should be provided.  

Reply: We understand the point of view raised by the respected reviewer. However, our main object of the manuscript is based on a qualitative study rather than a quantitative one.  We were so specific in taking the right and suitable samples for the study. In some cases, we repeated samples. However, many of them were made for the first time. Therefore, we did not consider the statistical information as a major part of the manuscript.

Reviewer 2 Report

Dear Authors,

I have read the manuscript entitled: New Insight on Photoisomerization Kinetics of Photo-switchable Thin Films Based on Azobenzene/Graphene Hybrid additives in Polyethylene Oxide and I think that the idea of this work is interesting  and results are worth to  be published, but this manuscript has to be corrected (especially Figure 1) and additionally, the minor revision should be done.

  • Figure 1. - The caption don’t go with the individual figures. We can see four figures (a, b, c, d) and five (a, b, c, d, e) in the caption under figures. In reality, figure (d) corresponds to the description: (e) the TGA curve for PEO-(MR–Graphene) hybrid composite thin film and next, the figure presented the HOMO and LUMO energy-band ….. is absent !!! So, I recommend to add this figure with comment in the manuscript or to remove this part from the caption under Figure 1.
  • Abstract – Three first sentences (lines 12-15) should be moved to the Introduction part and then Abstract may start: In this work we report a new insight………. Abstract is a brief summary of the studies presented in manuscript.
  • Misprints:

line 22: trans-cis (not ci);

line 33 trans-cis(not tans);

line 208: scenario, the (two spaces);

line 222: radicals generated (without space);

line 297: cis- to trans- isomerization (instead of cis- o trans-isomerization).

  • Line 58 – optical studies (instead of optical application).
  • Line 115 – should be absorption band (not vibrational band !!)
  • Line 148 and other, also in figures 1c, 2a, 4: Absorbance (Abs) is not in [%] only in arbitrary units [a.u.], because transmission (T) can be measured in [%] (0% - 100%) or can be depicted in the scale (0-1). Absorbance is simply defined as: Abs = log (1/T) where T is (0-1), so Abs [a.u].
  • Line 182: should be: “the photo-isomerization constant rate (p) as a function…” (instead of: “the photoisomerization constant rate,  as a function of time (p) ).

Summing up, I would like to let you know that in my opinion this manuscript can be published in the POLYMERS, but some, above mentioned, minor revision have to be done.

Author Response

Reviewer #2

Figure 1. - The caption doesn’t go with the individual figures. We can see four figures (a, b, c, d) and five (a, b, c, d, e) in the caption under figures. In reality, figure (d) corresponds to the description: (e) the TGA curve for PEO-(MR–Graphene) hybrid composite thin film, and next, the figure presented the HOMO and LUMO energy-band ….. is absent !!! So, I recommend to add this figure with a comment in the manuscript or to remove this part from the caption under Figure 1.

Reply: Authors removed this part from the caption as the reviewer suggested. Please see the revised version.

Abstract – Three first sentences (lines 12-15) should be moved to the Introduction part and then Abstract may start: In this work, we report a new insight………. The abstract is a brief summary of the studies presented in the manuscript.

Reply: Authors moved these sentences from the abstract to the introduction as the reviewer suggested. Please see the revised version, lines 16 – 19.

Misprints:

Reply: All misprints were corrected as the reviewer suggested, please see the revised version. 

Reviewer 3 Report

Authors deeply investigate the trans – cis isomerization of  AZO-PEO-Graphene composite showing its capability for many applications such as for many applications such as photochromic molecular switches or light-gated transistors. The manuscript is clearly written and data are clearly presented and discussed.

I have some minor comments:

1) I suggest to add FTIR spectra of PEO and graphene in figure 1b.

2) There are some typing error:

Line 22 ci-state

Line 33 tans

Line 104 ref [3]

Line 261 ref [32, 33]

Line 315 tans

Please check this sentence: The long-time isomerization (480 s of illumination is needed until a photo-stationary 164 equilibrium is achieved, compared with previous results reported [7, 9, 10]).

Author Response

Reviewer #3

1) I suggest adding FTIR spectra of PEO and graphene in figure 1b.

Reply: Authors added the FTIR for PEO and Graphene in Fig. 1b as the reviewer suggested. Please see Fig. 1b and page 3 lines 115-120 of the revised version.

2) There are some typing error: Line 22 ci-state, Line 33 tans, Line 104 ref [3], Line 261 ref [32, 33], Line 315 tans.

Reply: The authors corrected all the typing errors and revised the whole manuscript as the reviewer suggested. Please see the revised version taking into account that the line numbers have been shifted due to the addition of some paragraphs and statements based on the reviewer’s comments.

Please check this sentence: The long-time isomerization (480 s of illumination is needed until a photo-stationary 164 equilibrium is achieved, compared with previous results reported [7, 9, 10]).

Reply: Authors checked the sentence and correct it. Please see page 5, lines 175-176 of the revised version.

Round 2

Reviewer 1 Report

No more comments